

**Degradation Kinetics and Secondary Organic Aerosol Formation**
**from Eugenol by Hydroxyl Radicals**
Changgeng Liu[1,2], Yongchun Liu[1,3,4,a,*], Tianzeng Chen[1,4], Jun Liu[1,4], Hong He[1,3,4,*]
[1]State Key Joint Laboratory of Environment Simulation and Pollution Control,
Research Center for Eco-Environmental Sciences, Chinese Academy of Sciences,
Beijing 100085, China
[2]School of Biological and Chemical Engineering, Panzhihua University, Panzhihua
617000, China
[3]Center for Excellence in Regional Atmospheric Environment, Institute of Urban
Environment, Chinese Academy of Sciences, Xiamen 361021, China
[4]University of Chinese Academy of Sciences, Beijing 100049, China
[a]Currently at: Beijing Advanced Innovation Center for Soft Matter Science and
Engineering, Beijing University of Chemical Technology, Beijing 100029, China
*Correspondence to*: Yongchun Liu (liuyc@buct.edu.cn) and Hong He
(honghe@rcees.ac.cn)



**Abstract.** Methoxyphenols are an important organic component of wood-burning
emissions and considered to be potential precursors of secondary organic aerosols
(SOA). In this work, the rate constant and SOA formation potential for the OH-initiated
reaction of 4-allyl-2-methoxyphenol (eugenol) were investigated for the first time in an
oxidation flow reactor (OFR). The rate constant was $(8.01 \pm 0.40) \times 10^{-11}$ cm$^3$ molecule$^-$
$^1$ s$^{-1}$ as determined by the relative rate method. The SOA yield first increased and then
decreased as a function of OH exposure, and was also dependent on eugenol
concentration. The maximum SOA yields (0.11−0.31) obtained at different eugenol
concentrations could be expressed well by an one-product model. The carbon oxidation
state (OS$_C$) increased linearly and significantly as OH exposure rose, indicating that a
high oxidation degree was achieved for SOA. In addition, the presence of SO$_2$ (0−198
ppbv) and NO$_2$ (0−109 ppbv) was conducive to increasing SOA yield, for which the
maximum enhancement values were 38.57% and 19.17%, respectively. The N/C ratio
(0.032−0.043) indicated that NO$_2$ participated in the OH-initiated reaction,
subsequently forming organic nitrates. The results could be helpful for further
understanding the SOA formation potential from the atmospheric oxidation of
methoxyphenols and the atmospheric aging process of smoke plumes from biomass-
burning emissions.



## 1 Introduction

Wood combustion is a major contributor to atmospheric fine particulate matter (PM) (Bruns et al., 2016), which could contribute approximately 10−50% of the total organic fraction of atmospheric aerosols (Schauer and Cass, 2000). In some regions with cold climates, wood smoke-associated aerosols are estimated to account for more than 70% of $PM_{2.5}$ in winter (Jeong et al., 2008; Ward et al., 2006). Recently, significant potential of secondary organic aerosol (SOA) formation from wood smoke emissions has been reported (Bruns et al., 2016; Gilardoni et al., 2016; Tiitta et al., 2016; Ciarelli et al., 2017; Ding et al., 2017). In addition, the organic compounds derived from wood combustion and their oxidation products may contribute significantly to global warming due to their light-absorbing properties (Chen and Bond, 2010). It has been reported that wood smoke particles are predominant in the inhalable size range (Bari et al., 2010) and that their extracts are mutagenic (Kleindienst et al., 1986). Exposure to wood smoke can result in adverse health effects associated with acute respiratory infections, tuberculosis, lung cancer, cataracts, etc. (Bolling et al., 2009). Therefore, wood combustion has multifaceted impacts on climate, air quality, and human health.

Methoxyphenols produced by lignin pyrolysis are potential tracers for wood smoke, and their emission rates are in the range of 900−4200 mg kg$^{-1}$ fuel (Schauer et al., 2001; Simpson et al., 2005; Nolte et al., 2001). The highest level of methoxyphenols in the atmosphere always appears during a wood smoke-dominated period, with observed values up to several mg m$^{-3}$ (Schauer and Cass, 2000; Schauer et al., 2001; Simpson et





al., 2005). Methoxyphenols are semi-volatile aromatic compounds with low molecular
weight, and many of them are found to mainly exist in the gas phase at typical ambient
temperature (Simpson et al., 2005; Schauer et al., 2001). Thus, methoxyphenols can be
chemically transformed through gas-phase reactions with atmospheric oxidants (Coeur-
Tourneur et al., 2010a; Lauraguais et al., 2012, 2014a, 2014b, 2015, 2016; Yang et al.,
2016; Zhang et al., 2016; El Zein et al., 2015). The corresponding rate constants control
their effectiveness as stable tracers for wood combustion and atmospheric lifetimes. In
recent years, the rate constants for the gas-phase reactions of some methoxyphenols
with hydroxyl (OH) radicals (Coeur-Tourneur et al., 2010a; Lauraguais et al., 2012,
2014b, 2015), nitrate ($NO_3$) radicals (Lauraguais et al., 2016; Yang et al., 2016; Zhang
et al., 2016), chlorine atoms (Cl) (Lauraguais et al., 2014a) and ozone ($O_3$) (El Zein et
al., 2015) have been determined. Some studies have indicated significant SOA
formation from 2,6-dimethoxyphenol (syringol) and 2-methoxyphenol (guaiacol) with
respect to their reactions with OH radicals (Sun et al., 2010; Lauraguais et al., 2012,
2014b; Ahmad et al., 2017; Yee et al., 2013; Ofner et al., 2011). Although biomass-
burning emissions have been indicated to have great SOA formation potential via
atmospheric oxidation (Bruns et al., 2016; Gilardoni et al., 2016; Li et al., 2017; Ciarelli
et al., 2017; Ding et al., 2017), SOA formation and growth from methoxyphenols are
still poorly understood. Besides, the observed SOA levels in the atmosphere cannot be
well explained by the present knowledge on SOA formation, which reflects the fact that
a large number of precursors are not taken into account in the SOA-formation reactions



included in the atmospheric models (Lauraguais et al., 2012).

4-Allyl-2-methoxyphenol (eugenol), a type of methoxyphenols, is a typical

compound produced by ligin pyrolysis with a branched alkene group, and is widely
detected in the atmosphere (Schauer et al., 2001; Simpson et al., 2005; Bari et al., 2009).
Its average emission concentration and factor in beech burning are 0.032 μg m$^{-3}$ and
1.52 μg g$^{-1}$ PM, respectively, which are both higher than those (0.016 μg m$^{-3}$ and 0.762
μg g$^{-1}$ PM) of guaiacol (Bari et al., 2009). It has even be detected in human urine after
exposure to wood smoke (Dills et al., 2006). Eugenol has been observed to mainly
distribute in the gas phase in wood smoke emissions (Schauer et al., 2001), and its
gas/particle-partition coefficient is lower than 0.01 (Zhang et al., 2016), thus indicating
the importance of its gas-phase reactions in the atmosphere. For this reason, the aim of
this work was to determine the rate constant and explore the SOA formation potential
for eugenol in the gas-phase reaction with OH radicals using an Oxidation Flow Reactor
(OFR). In addition, the effects of $SO_2$ and $NO_2$ on SOA formation were investigated.
To our knowledge, this work represents the first determination of the rate constant and
SOA yield for the gas-phase reaction of eugenol with OH radicals.
**2   Experimental section**
The detailed schematic description of the experimental system used in this work is
shown in Figs. S1 and S2. The gas-phase reactions were conducted in the OFR, whose
detailed description has been presented elsewhere (Liu et al., 2014). Before entering
into the OFR, gas-phase species were mixed thoroughly in the mixing tube. The



reaction time in the OFR was 26.7 s, calculated according to the illuminated volume
(0.89 L) and the total flow rate (2 L min$^{-1}$). OH radicals were generated by photolysis
of $O_3$ in the presence of water vapor using a 254 nm UV lamp (Jelight Co., Inc.), and
their formation reactions have been described elsewhere (Zhang et al., 2017). The
concentration of OH radicals was governed by $O_3$ concentration and relative humidity
(RH). $O_3$ concentration was controlled by changing the unshaded length of a 185 nm
UV lamp (Jelight Co., Inc.). $O_3$ was produced by passing zero air through an $O_3$
generator (Model 610-220, Jelight Co., Inc.), and its concentration was in the range of
0.94−9.11 ppmv in this work measured with an $O_3$ analyzer (Model 205, 2B Technology
Inc.). RH and temperature in the OFR were (44.0 ±2.0)% and (301 ±1) K, respectively,
measured at the outlet of the OFR. The steady-state concentrations of OH radicals were
determined using $SO_2$ as the reference compound in separate calibration experiments.
It is a widely-used method for calculating OH exposure in the OFR, but could not well
describe the potential OH suppression caused by the added external OH reactivity
(Zhang et al., 2017; Lambe et al., 2015; Simonen et al., 2017; Li et al., 2015; Peng et
al., 2015, 2016). The decay of $SO_2$ from its reaction with OH radicals ($9 \times 10^{-13}$ cm$^3$
molecule$^{-1}$ s$^{-1}$) (Davis et al., 1979) was measured by a $SO_2$ analyzer (Model 43i, Thermo
Fisher Scientific Inc.). The concentration of OH radicals ([OH]) in this work ranged
from approximate $4.5 \times 10^9$ to $4.7 \times 10^{10}$ molecules cm$^{-3}$, and the corresponding OH
exposures were in the range of 1.21−12.55 $\times 10^{11}$ molecules cm$^{-3}$ s or approximate 0.93
to 9.68 d of equivalent atmospheric exposure.



An Aerodyne high-resolution time-of-flight aerosol mass spectrometer (HR-ToF-
AMS) was applied to perform online measurement of the chemical composition of
particles and the non-refractory submicron aerosol mass (DeCarlo et al., 2006). The
size distribution and concentration of particles were monitored by a scanning mobility
particle sizer (SMPS), consisting of a differential mobility analyzer (DMA) (Model
3082, TSI Inc.) and a condensation particle counter (CPC) (Model 3776, TSI Inc.).
Assuming that particles are spherical and non-porous, the average effective particle
density could be calculated to be 1.5 g cm$^{-3}$ using the equation $\rho = d_{va}/d_m$ (DeCarlo et al.,
2004), where $d_{va}$ is the mean vaccum aerodynamic diameter measured by HR-ToF-
AMS and $d_m$ is the mean volume-weighted mobility diameter measured by SMPS. The
mass concentration of particles measured by HR-ToF-AMS was corrected by SMPS
data in this work using the same method as Gordon et al. (2014). Eugenol and reference
compounds  were  measured  by  a  proton-transfer  reaction  time-of-flight  mass
spectrometer (PTR-QiToF-MS) (Ionicon Analytik GmbH). More experimental details
were described in the supplementary information.
**3   Results and discussion**
**3.1   Rate constant**
In order to investigate the possible photolysis of eugenol and reference compounds at
254 nm UV light in the OFR, the comparative experiments were conducted with UV
lamp turned on and turned off. The normalized mass spectra of eugenol and reference
compounds in the dark and light were shown in Fig. S3. The results showed that no



significant decay (<5%) by photolysis was observed and could be neglected. According
to the results reported by Peng et al. (2016), the photolysis of phenol and 1,3,5-
trimethylbenzene could be ignored when the ratio of exposure to 254 nm and OH is
lower than $1 \times 10^6$ cm s$^{-1}$, of which value in this work also met this condition. In addition,
the initial concentration of eugenol was determined with UV lamp turned on. Therefore,
the effect of photolysis could be neglected in this work.

The rate constant for the gas-phase reaction of eugenol with OH radicals was

determined by the relative rate method, which can be expressed as the following
equation (Coeur-Tourneur et al., 2010a; Yang et al., 2016; Zhang et al., 2016):
$\ln(C_{E0}/C_{Et}) = \ln(C_{R0}/C_{Rt})k_E / k_R$                    (1)
where $C_{E0}$ and $C_{Et}$ are the initial and real-time concentrations of eugenol, respectively.
$k_E$ is the rate constant of the eugenol reaction with OH radicals. $C_{R0}$ and $C_{Rt}$ are the
initial and real-time concentrations of reference compound, respectively. $k_R$ is the rate
constant of the reference compound with OH radicals, of which values for $m$-xylene
and 1,3,5-trimethylbenzene are $2.20 \times 10^{-11}$ and $5.67 \times 10^{-11}$ cm$^3$ molecule$^{-1}$ s$^{-1}$,
respectively (Kramp and Paulson, 1998; Coeur-Tourneur et al., 2010a).

Data obtained from the reactions were plotted in the form of Eq. (1) and were well

fitted by linear regression ($R^2 > 0.97$, Fig. 1). A summary of the slopes and the rate
constants are listed in Table 1. The errors in $k_E/k_R$ are the standard deviations generated
from the linear regression analysis and do not include the uncertainty in the rate
constants of the reference compounds. The rate constants are $(7.54 \pm 0.28) \times 10^{-11}$ and




$(8.47 \pm 0.51) \times 10^{-11}$ cm$^3$ molecule$^{-1}$ s$^{-1}$, respectively, when using 1,3,5-
trimethylbenzene and *m*-xylene as reference compounds. According to the US EPA
AOP WIN model based on the structure activity relationship (SAR) (US EPA, 2012)
the rate constant was calculated to be $6.50 \times 10^{-11}$ cm$^3$ molecule$^{-1}$ s$^{-1}$ (Table 1), which
is lower than that obtained in this work. Inaccurate performance of the AOP WIN model
has been observed for other multifunctional organics due to the inaccurate
representation of the eletronic effects of different functional groups on reactivity
(Coeur-Tourneur et al., 2010a; Lauraguais et al., 2012). This suggests that it is necessary
to determine the rate constants of multifunctional organics through lab experiments.
The rate constant determined in this work can be used to calculate the atmospheric
lifetime of eugenol with respect to its reaction with OH radicals. Assuming a typical
[OH] for a 24 h average value to be $1.5 \times 10^6$ molecules cm$^{-3}$ (Mao et al., 2009), the
corresponding lifetime of eugenol was calculated to be 2.31 h with the average rate
constant of $8.01 \times 10^{-11}$ cm$^3$ molecule$^{-1}$ s$^{-1}$. This short lifetime indicates that eugenol is
too reactive to be used as a tracer for wood smoke emissions, and also implies the
possible fast conversion of eugenol from gas-phase to secondary aerosol during the
transportation process.

The rate constant obtained in this work is about 2 orders of magnitude faster than

that for eugenol with NO$_3$ radicals ($1.6 \times 10^{-13}$ cm$^3$ molecule$^{-1}$ s$^{-1}$) (Zhang et al., 2016),
which suggests that the OH-initiated reaction of eugenol might be the main chemical
transformation in the atmosphere. The rate constants of the OH-initiated reactions of



guaiacol, 2,6-dimethylphenol, and syringol were $7.53 \times 10^{-11}$, $6.70 \times 10^{-11}$, and $9.66 \times$
$10^{-11}$ $cm^3$ $molecule^{-1}$ $s^{-1}$, respectively (Coeur-Tourneur et al., 2010a; Thuner et al., 2004;
Lauraguais et al., 2012). The reactivity of eugenol toward OH radicals is slightly higher
than those of guaiacol and 2,6-dimethylphenol, while slightly slower than that of
syringol. The presence of two methoxyl groups ($-OCH_3$) in syringol activates the
electrophilic addition of OH radicals to the benzene ring by donating electron density
through the resonance effect (Lauraguais et al., 2016). The activation effect of the
methoxyl group is much larger than those of alkyl groups (McMurry, 2004). In a recent
study, the reported energy barrier of $NO_3$ electrophilic addition to eugenol was about 2-
fold than that of 4-ethylguaiacol, indicating that the activation effect of the allyl group
($-CH_2CH=CH_2$) is lower than that of the ethyl group ($-CH_2CH_3$) (Zhang et al., 2016).
These results are in accordance with the activation effects of the substituants toward the
electrophilic addition of OH radicals (McMurry, 2004).
**3.2   Effects of eugenol concentration and OH exposure on SOA formation**
In this work, a series of experiments were conducted in the OFR with different eugenol
concentrations. The SOA yield was determined as the ratio of the SOA mass
concentration ($M_0$, $\mu g$ $m^{-3}$) to the reacted eugenol concentration ($\Delta$[eugenol], $\mu g$ $m^{-3}$)
(Kang et al., 2007). The experimental conditions and maximum SOA yields are listed
in Table 2. Fig. S4 shows the plots of the SOA yield versus OH exposure at different
eugenol concentrations. Higher concentrations resulted in higher amounts of
condensable products and subsequently increased SOA yield (Lauraguais et al., 2012).



SOA mass also directly influences the gas/particle partitioning, because SOA can serve
as the adsorption medium for oxidation products, and higher SOA mass generally
results in higher SOA yield (Lauraguais et al., 2012, 2014b). In the OFR, in all cases
the SOA yield first increased and then decreased as a function of OH exposure (Fig.
S4). This trend is the most common phenomenon observed in PAM reactor studies
(Lambe et al., 2015; Ortega et al., 2016; Simonen et al., 2017). In this work, according
to the $SO_2$ decay in the presence of eugenol and the OFR exposure estimator (v2.3)
developed by Jimenez's group based on the estimation equations reported in the
previous work (Li et al., 2015; Peng et al., 2015, 2016), the maximum reduction of OH
exposure by eugenol in the OFR was approximately 30%. Although OH suppression by
eugenol was not well determined in the OFR for the positive influence of $SO_2$ on SOA
formation, OH radicals were expected to be the main oxidant due to the fast reaction
rate constant of eugenol toward OH radicals obtained in this work. The decrease of
SOA yield at high OH exposure is possibly contributed from the C−C bond scission of
gas-phase species by further oxidation or heterogeneous reactions involving OH
radicals, which would generate a large amount of fragmented molecules that could not
condense on aerosol particles (Lambe et al., 2015; Ortega et al., 2016; Simonen et al.,

2017).

SOA yield can be described using a widely-used semi-empirical model on the basis

of the absorptive gas-particle partitioning of semi-volatile products, in which the overall
SOA yield (Y) is given by (Odum et al., 1996):



$$Y = \sum_i M_0 \frac{\alpha_i K_{om,i}}{1 + K_{om,i} M_0}$$ (2)
where $\alpha_i$ is the mass-based stoichiometric coefficient for the reaction producing the
semi-volatile product i, $K_{om,i}$ is the gas-particle partitioning equilibrium constant, and
$M_0$ is the total aerosol mass concentration.
The SOA yield data in Table 2 can be plotted in the form of Eq. (2) to obtain the
yield curve for eugenol (Fig. 2). The simulation of experimental data indicated that an
one-product model could accurately reproduce the data ($R^2 = 0.98$), while the use of
two or more products in the model did not significantly improve the fitting quality.
Odum et al. (1996) reported that the SOA yield data from the oxidation of aromatic
compounds could be fitted well using a two-product model. However, an one-product
model was also efficient for describing the SOA yields from the oxidation of aromatics
including methoxyphenols (Lauraguais et al., 2012, 2014b; Coeur-Tourneur et al.,
2010b). The success of simulation with an one-product model in this work is likely to
indicate that the products in SOA have similar values of $\alpha_i$ and $K_{om,i}$, i.e., that the
obtained $\alpha_i$ (0.36 ±0.02) and $K_{om,i}$ (0.013 ±0.002 $m^3$ $ug^{-1}$) represent the average values.
In this work, considering that the composition of SOA was not determined, the volatility
basis set (VBS) approach was not applied to simulate SOA yields. Fig. S5 shows a plot
of the SOA mass concentration ($M_0$) versus the reacted eugenol concentration
(Δ[eugenol]). Its slope was 0.37 as obtained using linear least-squares fitting, which is
very close to the $\alpha_i$ value (0.36). This suggests that the low-volatility products formed
in the reaction almost completely disperse on the particle phase according to the



theoretical partition model (Lauraguais et al., 2012, 2014b). In other words, SOA yield
was approximately an upper limit for eugenol oxidation in the OFR. In view of the
residence time in this work, it seems be in contradiction with the recommendation of
longer residence time made by Ahlberg et al. (2017), who found that the condensation
of low-volatility species on SOA in the OFR was often kinetically limited at low mass
concentrations. In our recent experiments (not published), the SOA yields for guaiacol
oxidation by OH radicals obtained under the similar experimental conditions as this
work, could be comparable to those obtained in the chamber studies (Fig. S6). This
suggests that the effect of kinetic limitations on SOA condensation for the OH-initiated
oxidation of methoxyphenols in this system might be not important.

Elemental ratios (H/C and O/C) could provide insights into SOA composition and

chemical processes along with aging (Bruns et al., 2015). As shown in Fig. 3, O/C ratio
of SOA increases and H/C ratio decreases with increasing OH exposure, because
oxygen-containing functional groups are formed in the oxidation products. In addition,
the organic mass fractions of m/z 44 ($CO_2^+$) and m/z 43 (mostly $C_2H_3O^+$), named $f_{44}$
and $f_{43}$, respectively, could also provide information about the nature of SOA formation.
Fig. S7 shows the evolution of $f_{44}$ and $f_{43}$ versus OH exposure at low (272 µg m$^{-3}$) and
high (1328 µg m$^{-3}$) concentrations of eugenol. The values of $f_{44}$ were much higher than
those of $f_{43}$, and increased significantly as a function of OH exposure, suggesting that
SOA formed in the experiments became more oxidized. The $f_{44}$ value in this work
ranges up to 0.26, which is consistent with that observed for ambient low-volatility (LV-





OA), higher than 0.25 (Ng et al., 2010).
The average carbon oxidation state ($OS_C$) proposed by Kroll et al. (2011) is
considered a more accurate indicator of the oxidation degree of atmospheric organic
species than the O/C ratio alone, because it takes into account the saturation level of the
carbon atoms in the SOA. $OS_C$ is defined as $OS_C = 2O/C - H/C$ (Kroll et al., 2011),
calculated according to the elemental composition of SOA measured by HR-ToF-AMS.
In this work, the $OS_C$ values obtained at low (272 µg m$^{-3}$) and high (1328 µg m$^{-3}$)
concentrations of eugenol were compared. As shown in Fig. 3, $OS_C$ values for low
concentration (0.035−1.78) were much larger than those for high concentration
(0.0036−1.09), and increased linearly ($R^2 > 0.96$) with OH exposure of (1.21−12.55) ×
$10^{11}$ molecules cm$^{-3}$ s. The results are well supported by the evolution of SOA mass
spectra obtained by HR-ToF-AMS at the same eugenol concentrations (Fig. S8).
Similar trends have been observed in the smog chamber and PAM reactor (Simonen et
al., 2017; Ortega et al., 2016). The $OS_C$ value in this work extends as high as 1.78,
which is in good agreement with that observed for ambient LV-OA, up to 1.9 (Kroll et
al., 2011). Recently, Ortega et al. (2016) reported that the $OS_C$ value for SOA formed
from ambient air in an OFR ranged up to 2.0; and Simonen et al. (2017) determined a
high $OS_C$ value (> 1.1) for SOA formed from the OH-initiated reaction of toluene in a
PAM reactor with an OH exposure of 1.2 ×$10^{12}$ molecules cm$^{-3}$ s. In general, the $OS_C$
values for the PAM reactor are higher than those for smog chambers due to the high
OH exposure in the PAM reactor (Simonen et al., 2017; Ortega et al., 2016; Lambe et





al., 2015). Higher $OS_C$ value indicates greater age, where the SOA components are
further oxidized through heterogeneous oxidation, adding substantial oxygen and
reducing hydrogen in the molecules in the particle-phase to increase $OS_C$ values despite
the overall loss of SOA mass (Ortega et al., 2016).
**3.3   Effect of $SO_2$ on SOA formation**
As shown in Fig. 4, the presence of $SO_2$ favored SOA formation, and the sulfate
concentration increased linearly ($R^2 = 0.99$) as a function of OH exposure. The
maximum SOA yield enhancement of 38.57% was obtained at OH exposure of 5.41 $\times$
$10^{11}$ molecules $cm^{-3}$ s, and then decreased with the increase of OH exposure due to the
fragmented molecules formed through the oxidation of gas-phase species by high OH
exposure (Lambe et al., 2015; Ortega et al., 2016; Simonen et al., 2017). The SOA yield
and sulfate concentration both increased linearly ($R^2 > 0.97$) as $SO_2$ concentration
increased from 0 to 198 ppbv at OH exposure of 1.21 $\times 10^{11}$ molecules $cm^{-3}$ s (Fig. S9).
Compared to the initial SOA yield (0.049) obtained in the absence of $SO_2$, the SOA
yield (0.066) obtained in the presence of 198 ppbv $SO_2$ was enhanced by 34.69%. In
previous studies, Kleindienst et al. (2006) reported that the SOA yield from $\alpha$-pinene
photooxidation increased by 40% in the presence of 252 ppbv $SO_2$; Liu et al. (2016b)
recently found that the SOA yield from 5 h photochemical aging of gasoline vehicle
exhaust was enhanced by 60−200% in the presence of ~150 ppbv $SO_2$.

As shown in Figs. 4 and S7, the increase of sulfate concentration was favorable for

SOA formation. In this system, it is difficult to completely remove trace $NH_3$, thus the





formed sulfate was the mixture of sulfuric acid ($H_2SO_4$) and a small amount of
ammonium sulfate (($NH_4$)$_2SO_4$). The in situ particle acidity was calculated as the $H^+$
concentration ([$H^+$], 40.23−648.39 nmol m$^{-3}$) according to the AIM-II model for the H−
$NH_4^+ - SO_4^{2-} - NO_3^- - H_2O$ systems (http://www.aim.env.uea.ac.uk/aim/model2 /model2
a.php; Liu et al., 2016b). The detailed description of the calculation method has been
represented elsewhere (Liu et al., 2016b). The elevated concentration of sulfate in the
particle phase with the increases of $SO_2$ concentration and OH exposure is an important
reason for the enhanced SOA yields (Kleindienst et al., 2006; Liu et al., 2016b). Cao
and Jang (2007) indicated that SOA yields from the oxidation of toluene and 1,3,5-
trimethylbenzene increased by 14−36% in the presence of acid seeds, with [$H^+$] of
240−860 nmol m$^{-3}$ compared to those obtained in the presence of nonacid seeds. Similar
results concerning the effect of particle acidity on SOA yields were reported in other
studies (Kleindienst et al., 2006; Liu et al., 2016b; Jaoui et al., 2008; Xu et al., 2016).
However, Ng et al. (2007b) found that particle acidity had a negligible effect on SOA
yields from photooxidation of aromatics, possibly due to the low RH (~5%) used in
their work. The water content of aerosol plays an essential role in acidity effects (Cao
and Jang, 2007). Under acidic conditions, the gas-phase oxidation products of eugenol
would be partitioned more quickly into the particle-phase and further oxidized into low
volatility products, or produce oligomeric organics by acid-catalyzed heterogeneous
reactions, subsequently enhancing SOA yields (Cao and Jang, 2007; Jaoui et al., 2008;
Liu et al., 2016b; Xu et al., 2016). In addition, the formed sulfate not only serves as the





substrate for product condensation and likely participates in new particle formation
(NPF) (Jaoui et al., 2008; Wang et al., 2016), but also enhances the surface areas of
particles to facilitate heterogeneous reactions on aerosols (Xu et al., 2016). These roles
of sulfate are also favorable for increasing SOA yields. Recently, Friedman et al. (2016)
have indicated that $SO_2$ could participate in the oxidation reactions of α-and β-pinene
and perturbs their oxidation in the OFR, but this possible effect could be ignored in this
work due to the relatively high RH and the negligible S/C ratio observed by HR-ToF-
AMS (Friedman et al., 2016).

**3.4  Effect of $NO_2$ on SOA formation**

It is well known that high $NO_x$ concentration almost always plays a negative role in
NPF and SOA formation because the reaction of NO with $RO_2$ radicals results in the
formation of more volatile products compared to the reaction of $HO_2$ with $RO_2$ radicals
(Sarrafzadeh et al., 2016). Previous studies reported that nitro-substituted products were
the main products for SOA formed from OH-initiated reactions of phenol precursors
including methoxyphenols, in the presence of $NO_x$ (Finewax et al., 2018; Ahmad et al.,
2017; Lauraguais et al., 2012, 2014b). Thus, the effect of $NO_2$ on SOA formation from
eugenol oxidation by OH radicals was investigated. As shown in Fig. 5, the nitrate
concentration measured by HR-ToF-AMS increased as a function of OH exposure in
the presence of 40 ppbv $NO_2$, but it was much lower than the sulfate concentration (Fig.
4) even though the OH rate constant for $NO_2$ was faster than that for $SO_2$ (Davis et al.,
1979; Atkinson et al., 1976). The possible explanation is that the formed $HNO_3$ mainly



exists in the gas phase, and the relatively high temperature (301 ±1 K) is not favorable
for gaseous $HNO_3$ distribution in the particle phase (Wang et al., 2016). It has been
indicated that the temperature range for the greatest loss of nitrate is 293−298 K (Keck
and Wittmaack, 2005). As illustrated in Fig. 5, the SOA yield enhancement and N/C
ratio both increased firstly and then decreased with rising OH exposure. The increase
of $NO_2$ concentration (40−109 ppbv) is beneficial to SOA yields (0.053−0.062), N/C
ratio (0.032−0.041), and nitrate formation (4.29−6.30 µg m$^{-3}$) (Fig. S10). Compared to
the presence of 41 ppbv $SO_2$, the maximum SOA yield enhancement (19.17%) in the
presence of 40 ppbv $NO_2$ is lower. For most aromatic precursors, the addition of ppbv
levels of $NO_2$ should have a negligible effect on SOA formation, because the rate
constants of OH-aromatic adducts with $O_2$ and $NO_2$ are on the order of approximate $10^{-16}$
and $10^{-11}$ $cm^3$ molecule$^{-1}$ s$^{-1}$, respectively (Atkinson and Arey, 2003). But, for phenol
precursors only about 0.5 ppbv $NO_2$ is enough to compete with $O_2$ for the reaction with
OH-aromatic adducts (Finewax et al., 2018). Therefore, the enhancement effect of $NO_2$
on SOA formation might be relevant to the special case of phenols or methoxyphenols
but not other aromatic precursors.
It is noteworthy that the N/C ratio is in the range of 0.032−0.043, suggesting that
$NO_2$ participated in the OH reaction of eugenol, through the addition to the OH-eugenol
adduct (Peng and Jimenez, 2017). Recently, Hunter et al. (2014) found that $NO_2$
participated in the OH reactions of yclic alkanes, and the N/C ratios were in the range
of 0.031−0.064, higher than those obtained in this work. The nitro-substituted products





are reported to be the main reaction products of the OH reactions of guaiacol and
syringol in the presence of $NO_2$ (Lauraguais et al., 2014b; Ahmad et al., 2017). The N-
containing products might be also formed through the reactions involving with $NO_3$
radicals, which are possibly generated by the reaction between $NO_2$ and $O_3$ in this
system (Atkinson, 1991). But, the specific contribution of $NO_3$ radicals could not be
quantified in this work. The relative low volatility of these products could reasonably
contribute to SOA formation (Duportéet al., 2016; Liu et al., 2016a). In addition, higher
$NO_2$/NO ratio favors the formation of nitro-substituted products, which are potentially
involved in NPF and SOA growth (Pereira et al., 2015). Ng et al. (2007a) also indicated
that $NO_x$ could be beneficial to SOA formation for sesquiterpenes, due to the formation
of low volatility organic nitrates and the isomerization of large alkoxy radicals, resulting
in less volatile products. The decrease in the N/C ratio at high OH exposure suggested
that more volatile products were generated through the oxidation of particle-phase
species by OH radicasls.
The $NO^+ / NO_2^+$ ratios measured by HR-ToF-MS are widely used to identify
inorganic and organic nitrates. The $NO^+ / NO_2^+$ ratios for inorganic nitrates have been
reported to be in the range of 1.08−2.81 (Farmer et al., 2010; Sato et al., 2010). The
ratio ranged from 2.06 to 2.54 in this work as determined by HR-ToF-AMS using
ammonium nitrate as the calibration sample. However, the $NO^+ / NO_2^+$ ratios during
oxidation of eugenol in the presence of 40 ppbv $NO_2$ were 3.98−6.09. They were higher
than those for inorganic nitrates and consistent with those for organic nitrates



(3.82−5.84) from the photooxidation of aromatics (Sato et al., 2010). The abundance of
organic nitrates could be estimated from the N/C ratios determined in this work.
Assuming that the oxidation products in the SOA retain 10 carbon atoms, the yields of
organic nitrates are in the range of 32−43%, which are comparable to those reported in
earlier studies (Liu et al., 2015; Hunter et al., 2014). Liu et al. (2015) reported that the
nitrogen-containing organic mass contributed $31.5 \pm 4.4\%$ to the total SOA derived
from m-xylene oxidation by OH radicals. Hunter et al. (2014) estimated the organic
nitrate yields of SOA to be 31−64%, formed in the OH-initiated reactions of acyclic,
monocyclic, and polycyclic alkanes. This range obtained in this work should be the
upper limit due to the possibility of C−C bond scission of gas- and particle-phase
organics oxidized by high OH exposure. Besides, the maximum yield of nitrates for a
single reaction step is expected to be approximately 30% (Ziemann and Atkinson, 2012),
this suggests that multiple reaction steps are needed.
**3.5   Atmospheric implications**
Biomass burning not only serves as a major contributor of atmospheric POA, but also
has great SOA formation potential through atmospheric oxidation (Bruns et al., 2016;
Gilardoni et al., 2016; Li et al., 2017; Ciarelli et al., 2017; Ding et al., 2017). Recent
studies have indicated that SOA formed from biomass burning plays an important role
in haze pollution in China (Li et al., 2017; Ding et al., 2017). Residential combustion
(mainly wood burning) could contribute approximately 60−70% to SOA formation in
winter at the European scale (Ciarelli et al., 2017). In addition, methoxyphenols are the



major component of OA from biomass burning (Bruns et al., 2016; Schauer and Cass,
2000). Based on our results and those of previous studies (Sun et al., 2010; Lauraguais
et al., 2012, 2014b; Ahmad et al., 2017; Yee et al., 2013; Ofner et al., 2011), it should
pay more attenion on the SOA formation from the OH oxidation of biomass burning
emissions and its subsequent effect on haze evolution, especially in China with
nationwide biomass burning and high daytime average [OH] in the ambient atmosphere
$((5.2-7.5) \times 10^6$ molecules $cm^{-3}$) (Yang et al., 2017). Meanwhile, the potential
contributions of $SO_2$ and $NO_2$ to SOA formation should also be taken into account,
because the concentrations of $NO_x$ and $SO_2$ could be up to close 200 ppbv in the
severely polluted atmosphere in China (Li et al., 2017). Although eugenol
concentrations in this work are higher than those in the ambient atmosphere, the results
obtained in this work could provide new information for SOA formation from the
atmospheric oxidation of methoxyphenols, and might be useful for SOA modeling,
especially for air quality simulation modeling of the specific regions experiencing
serious pollution caused by fine particulate matter.

N-containing products formed from the oxidation of methoxyphenols could

contribute to water-soluble organics in SOA (Lauraguais et al., 2014b; Yang et al., 2016;
Zhang et al., 2016), which have been widely detected in atmospheric humic-like
substances (HULIS) (Wang et al., 2017). Due to their surface-active and UV-light-
absorbing properties, HULIS could influence the formation of cloud condensation
nuclei (CCN), solar radiation balance, and photochemical processes in the atmosphere




(Wang et al., 2017). The high reactivity of methoxyphenols toward atmospheric radicals
suggests that SOA was formed from their oxidation processes with relatively high
oxidation level, subsequently leading to SOA with strong optical absorption and
hygroscopic properties (Lambe et al., 2013; Massoli et al., 2010). Therefore, SOA
formed from the reactions of methoxyphenols with atmospheric oxidants might have
important effects on air quality and climate. In addition, the experimental results from
this study would help to further the understanding of the atmospheric aging process of
smoke plumes from biomass-burning emissions.
**4  Conclusions**
For the first time, the rate constant and SOA foramtion for the gas-phase reaction of
eugenol with OH radicals were investigated in an OFR. The second-order rate constant
of eugenol with OH radicals was $(8.01 \pm 0.40) \times 10^{-11}$ cm$^3$ molecule$^{-1}$ s$^{-1}$, measured by
the relative rate method, and the corresponding atmospheric lifetime was 2.31 h. In
addition, the significant SOA formation of eugenol oxidation by OH radicals was
observed. The maximum SOA yields (0.11−0.31) obtained at different eugenol
concentrations could be expressed well by an one-product model. SOA yield was
dependent on OH exposure and eugenol concentration, which firstly increased and then
decreased as a function of OH exposure due to the possible C−C bond scission of gas-
phase species by further oxidation or heterogeneous reactions involving OH radicals.
The OS$_C$ and O/C ratio both increased significantly as a function of OH exposure,
suggesting that SOA became more oxidized. The presence of SO$_2$ and NO$_2$ was helpful





to increase SOA yield, and the maximum enhanced yields were 38.57% and 19.17%,
respectively. The observed N/C ratio of SOA was in the range of 0.032−0.043,
indicating that $NO_2$ participated in the OH-initiated reaction of eugenol, consequently
producing organic nitrates. The experimental results might be helpful to further
understand the atmospheric chemical behavior of eugenol and its SOA formation
potential from OH oxidation in the atmosphere.
**Data availability**
The experimental data are available upon request to the corresponding authors.
**Competing interests**
The authors declare that they have no conflict of interest.
**Aknowledgements**
This work was financially supported by the National Key R&D Program of China
(2016YFC0202700), the National Natural Science Foundation of China (21607088),
China Postdoctoral Science Foundation funded project (2017M620071), and the
Applied Basic Research Project of Science and Technology Department of Sichuan
Province (2018JY0303). Liu Y. would like to thank Beijing University of Chemical
Technology for financial supporting. Authors would also acknowledge the
experimental help provided by Dr. Xiaolei Bao from Hebei Provincial Academy of
Environmental Sciences, Shijiazhuang, China.





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





**Table 1.** Rate constant for gas-phase reaction of eugenol with OH radicals and
associated atmospheric lifetime.

| Compound | Structure | References | $k_E/k_R$ | $k_E$ [a] | $k_{OH}$ [a] | $k_E$ (average) [a] | $\tau_{OH}$ (h)[b] |
|---|---|---|---|---|---|---|---|
| eugenol | | 1,3,5-trimethylbeneze | 1.33 ±0.05 | 7.54 ±0.28 | | | |
| ($C_{10}H_{12}O_2$) | | $m$-xylene | 3.85 ±0.23 | 8.47 ±0.51 | 6.50[c] | 8.01 ±0.40 | 2.31 |

[a] Units of $10^{-11}$ cm$^3$ molecule$^{-1}$ s$^{-1}$.
[b] Atmospheric lifetime in hours. $\tau_{OH}=1/k_E[OH]$, assuming a 24 h average [OH] = 1.5 ×
$10^6$ molecules cm$^{-3}$ (Mao et al., 2009).
[c] Calculated using US EPA AOP WIN model (US EPA, 2012).
**Table 2.** Experimental conditions and results.

| Expt. | [eugenol]$_0$[a] (µg m$^{-3}$) | Δ[eugenol][b] (µg m$^{-3}$) | $M_0$[c] (µg m$^{-3}$) | $Y_{max}$[d] | OH Exposure[e] ($10^{11}$ molecules cm$^{-3}$ s) | $\tau$[f] (d) |
|---|---|---|---|---|---|---|
| #1 | 272 | 265 | 29 | 0.11 | 5.41 | 4.17 |
| #2 | 351 | 339 | 54 | 0.16 | 5.41 | 4.17 |
| #3 | 485 | 474 | 83 | 0.18 | 5.41 | 4.17 |
| #4 | 636 | 625 | 145 | 0.23 | 5.41 | 4.17 |
| #5 | 874 | 858 | 241 | 0.28 | 7.37 | 5.68 |
| #6 | 1327 | 1304 | 399 | 0.31 | 8.91 | 6.87 |

[a] Initial eugenol concentrations.
[b] Reacted eugenol concentrations.
[c] SOA concentrations.
[d] Maximum SOA yields.
[e] Corresponding OH exposure of maximum SOA yields.
[f] Corresponding atmospheric aging time of maximum SOA yields, calculated using a
typical [OH] in the atmosphere in this work (1.5 × $10^6$ molecules cm$^{-3}$) (Mao et al.,

823 2009).

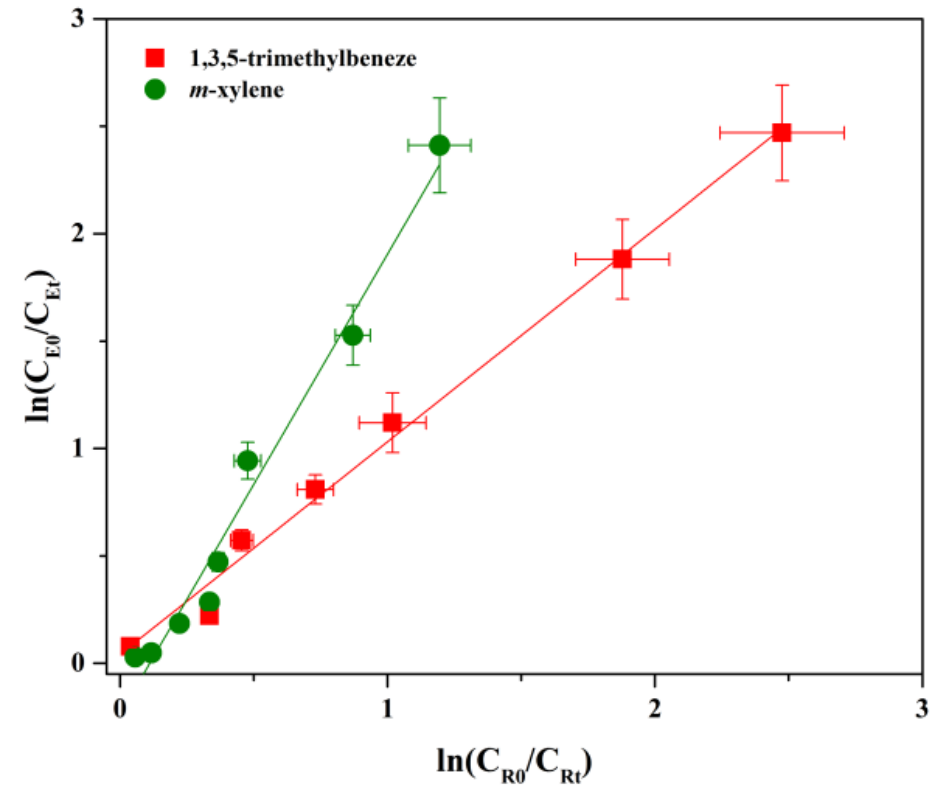

**Figure 1.** Relative rate plots for gas-phase reaction of OH radicals with eugenol.

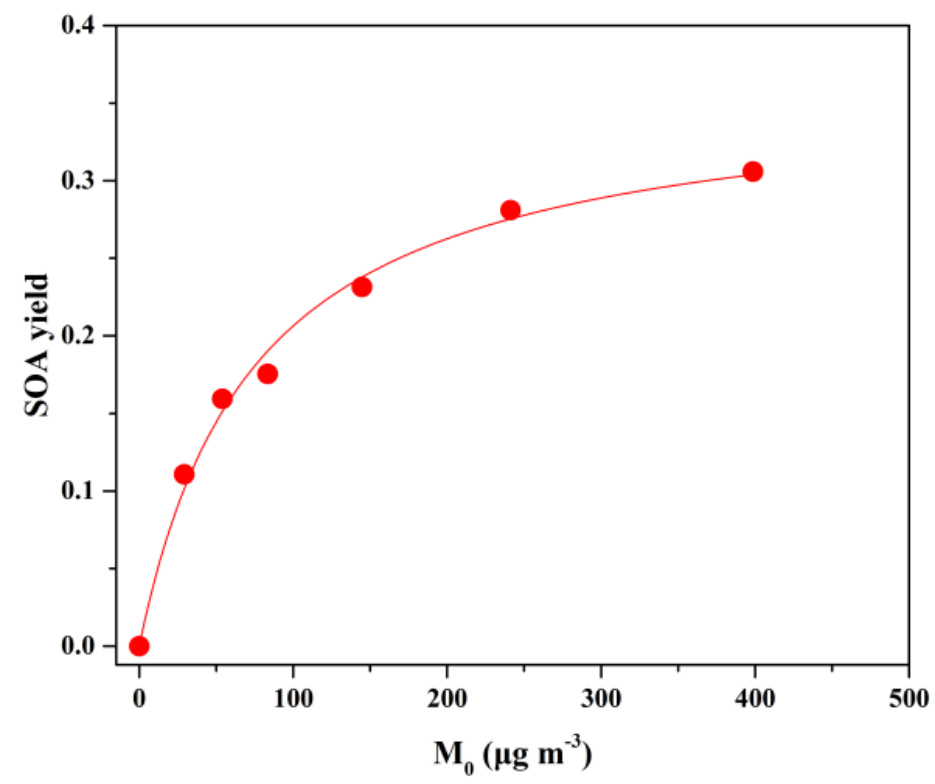

**Figure 2.** Maximum SOA yield as a function of SOA mass concentration ($M_0$) formed from the OH reactions at different eugenol concentrations. The solid line was fit to the experimental data using an one-product model. Values for $\alpha_i$ and $K_{om,i}$ used to generate the solid line are (0.36 ± 0.02) and (0.013 ± 0.002), respectively.

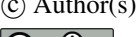



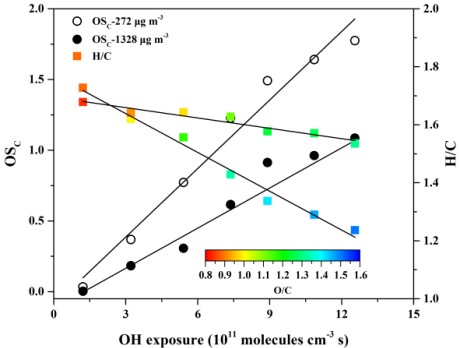

**Figure 3.** $OS_C$, H/C, and O/C vs. the OH exposure for SOA formed at two eugenol

concentrations (272 and 1328 μg m$^{-3}$).

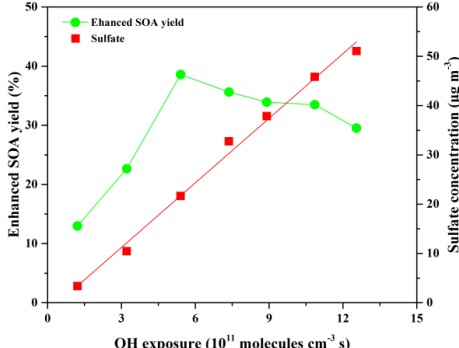

**Figure 4.** Evolution of the enhanced SOA yield and sulfate formation as a function of

OH exposure in the presence of 41 ppbv $SO_2$ at average eugenol concentration of 273

μg m$^{-3}$.



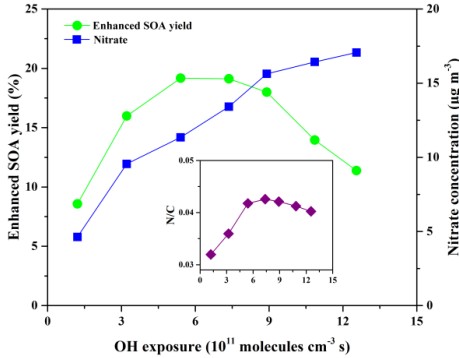


**Figure 5.** Evolution of the enhanced SOA yields, nitrate formation, and N/C ratioas a
function of OH exposure in the presence of 40 ppbv NO$_2$ at average eugenol
concentration of 273 μg m$^{-3}$.