# Peer review of "gas-phase reaction of eugenol with hydroxyl radicals"

_Atmospheric Chemistry and Physics, 2018_

## Referee Comment (RC1) · Anonymous Referee #1 · 18 Nov 2018

General comments:

In this manuscript, the authors report the first measurements of the rate constant for the reaction of eugenol, an atmospherically abundant methoxyphenol, with the hydroxyl radical in the gas phase. The results are placed in the context of other previously investigated methoxyphenols, including a helpful discussion of substituent effects. The authors also present a detailed characterization of the SOA yield and its response to SO2 and NO2, including a surprising enhancement in SOA yield due to the presence of NO2, which may apply to other methoxyphenols. The experimental work is thorough and precise, with appropriate controls, and the manuscript is well organized. I

recommend the manuscript for publication following minor revisions.

Specific comments:

144: Though photolysis does not contribute to the decay of eugenol, could it contribute to the evolution of the oxidation products, either in the gas or particle phases? Though it is not a focus of this study (one can imagine forming SOA in one OFR, scrubbing any remaining ozone, and then irradiating the products in a second OFR), perhaps the possibility of photolysis of the oxidation products should be acknowledged at the end of this paragraph.

164: I enjoyed the comparison of the experimental and AOP WIN-predicted rate constants. I wonder if the comparison should be placed in the context of other methods of prediction. For example, the DFT-predicted rate constant for the reaction of guaiacol with the hydroxyl radical (DOI: 10.1002/poc.3713) is about 1.6 times greater than the experimental value (DOI: 10.1021/jp1071023). In this context, the present agreement, with a predicted value about 0.8 times the experimental value, seems quite good.

252: The lack of kinetic limitation to condensation is very interesting. Could this observation be related broadly to the viscosity of the SOA derived from eugenol and guaiacol? How does the present relative humidity of about 44% compare to that in the previous OFR and smog chamber experiments discussed in the comparisons?

365: Perhaps the detailed discussion of the effects of NO2 on the SOA yield would benefit from a mechanistic schematic.

427: I agree that it is very likely that the SOA derived from eugenol is light-absorbing. Other products, in addition to those containing nitrogen, could conceivably contribute to the proposed absorptivity. For example, products of oligomerization like biphenyls have been observed in the aqueous oxidation of phenolic species (DOI:10.5194/acp-14-13801-2014), and this pathway is likely relevant in the highly-concentrated aerosol phase.

[Figure]

Technical corrections:

77: In the phrase "a type of methoxyphenols", methoxyphenol should be singular. Perhaps this phrase is redundant and could be omitted.

115-116: Two instances of "approximate" should be "approximately".

126: "Vaccum" should be "vacuum".

199: In the caption to Figure S4, perhaps explain that the arrows indicate the maximum values (i.e, those listed in Table 2).

217-218: This phrase should be reworded to better reflect that the fragmentation occurs in the particles and that the products subsequently volatilize out of the particles.

238: This phrase is slightly confusing, since some measure of composition is determined using the AMS.

368: Should be "cyclic".

383: Should be "radicals".

414-415: This phrase should be reworded to give, for example, "more attention should be paid to SOA formation..."

442: Should be "formation".

---

## Referee Comment (RC2) · Anonymous Referee #2 · 3 Dec 2018

This study investigated the OH reaction rate of pure eugenol compound and its SOA yield with a custom-built oxidation flow reactor (OFR). The impact of NO2 and SO2 influence on SOA formation was also investigated. The results of the study might be very interesting to many, yet quite a few items need to be clarified before it can be accepted for publication.

Experiment design: The manuscript mainly deals with two topics: the rate constant of Eugenol-OH reaction and SOA yield of Eugenol. As for the 6 experiments listed in the Table 1, apart form the Eugenol, how about the combination of other species such like m-xylene/1,3,5-TMB/SO2/NO2. For experiments determines the rate constant for the

reaction with OH, were SO2/NO2 added? And for SOA yield studies, what was the reference compound to derive OH? It seemed that here SO2/m-xylene/1,3,5-TMB are all not appropriate to sever as the reference compound since m-xylene/1,3,5-TMB they themselves are also SOA precursors and SO2 instead influence the SOA yield. It is quite confusing. The authors need to clarify in Table 1 what combinations (including reference compound for deriving OH) of species are prepared for obtaining the rate constant, and what combinations instead are for studying the SOA yield.

From the study only the overall rate constant for reaction with OH was obtained. It seemed that not so much degradation kinetics are presented as indicated by the title.

Line 28: The enhancement values need not to have 4 significant figures. Lines 77-84: What's the overall concentration of eugenol in ambient air? How important does this precursor compared with other Methoxyphenols. Line 105: The [O3] was in the range of 0.94-9.11 ppmv while you want to explore the reaction between OH radical and eugenol. Can you clarify whether such high level of O3 was just used to generate OH radical in the mixing tube, or they indeed existed in the flow tube? If it is the latter case as your supplement material shows, then an evaluation of the interference from the O3 is needed. Lines 109-110: This is not acceptable. The OH reactivity applied in this experiment is at least 80 s-1 to 380 s-1 with only Eugenol accounted (using OH reaction rate calculated in section 3.1), not mentioned the SO2 (0-198ppb) and NO2 (0-109ppb in line 27) added in the later experiment. It suggested the calculated OH exposure should be at least several times lower than the OH exposure calculated without considering the external OH reactivity (=0 s-1 assumed in original calculation) (Peng et al., 2015) Lines 116-117: Better add the assumed average [OH] and the reference as well. Lines 124-125: How do the authors calibrate their aerodynamic size distribution in AMS? If the authors consider the chemical-composition based particle density (Kuwata et al., 2012; Salcedo et al., 2006), how about it when compared to the effective aerosol density applied here? The aerosol size distribution of each experiment should be considered separately. The 100% full cut size of AMS lens is around 600

nm in mobility size (Nault et al., 2018). The effective density calculation from dva/dm could be biased if the aerosol in OFR grow beyond the AMS lens cut. Lines 152-153: The decay of SO2 was used to calculate [OH], so why was SO2 not used as the reference compound in the relative rate method? As mentioned above, the reader should be informed in Table 1 what were initially added and what are the reference compound. Lines 135-137ïijŽThis is confusing. Did the authors examine a full cycle of UV light applied in the experiment? Lines 140-141: What is the photon flux of 254 nm in OFR. How do the authors determine 254 exposure/OH exposure ratio? Lines 163-164: Is it possible that the difference between your measurement and the theory was caused by the O3 reaction? Lines 164-167: Have the authors considered the potential wall loss of three species, which could result in different species decay ratios. Thus extra uncertainty on OH reaction rate coefficient of Eugenol could be introduced. Lines 181-183: How about those reaction rate coefficients estimated from experiment when compared to those from the SAR method? Line 207: The decrease have also been reported in references of (Palm et al., 2016; Palm et al., 2018) Line 211: Should be larger than 30% based on Fig. S4 in (Peng et al., 2015). Please show the detailed calculation results. Line 227: How about the wall loss of aerosols in the flow tube. The authors could examine the wall loss by directly injecting aerosols into the OFR. Line 263: Have the author consider the NH4NO3→CO2 effect, which could influence fCO2 and thus O/C and H/C ratio substantially. This bias could be larger especially in the later NO2+ Eugenol experiment. Line 284-286: For saying this, OH exposure should be compared. Line 294-296: We cannot definitely conclude that the decrease is due to the fragmented molecules formed through the oxidation of gas-phase species. So better add "probably" or "possibly" before "due to". Line 311: Can the formed SO4 be fully explained by the SO2 decay in SO2+ Eugenol experiment? Line 323: Why does the eugenol can partition quickly under acidic aerosol condition? Lines 360-362: In the paper of Finewax et al. (2018), it is phenoxy radical rather than OH-aromatic adducts that react with NO2 or O2. In fact, the formations of phenoxy radical and OH-aromatic adduct from phenols are totally different in reaction pathways. Lines 366-367: Still,

the reaction pathway through the NO2 addition on the phenoxy radical was neglected by the author. Line 375: The authors could still specify the NO3 exposure compared to OH exposure by assuming thermo-steady state of NO2 and O3. Line 391: In this method, if the NO and NO2+ ions from organonitrate were missed, the organic nitrate calculation was underestimated (Farmer et al., 2010). The authors can use the real-time NO+/NO2+, and NO+/NO2+ ratio from NH4NO3 and organonitrate (a factor of 2.25 ) (Fry et al., 2018) to calculate -ONO2 group concentration for organonitrates. Line 839: "ratioas" should be "ratio as".

References: Salcedo, D., Onasch, T. B., Dzepina, K., Canagaratna, M. R., Zhang, Q., Huffman, J. A., DeCarlo, P. F., Jayne, J. T., Mortimer, P., Worsnop, D. R., Kolb, C. E., Johnson, K. S., Zuberi, B., Marr, L. C., Volkamer, R., Molina, L. T., Molina, M. J., Cardenas, B., Bernabe, R. M., Marquez, C., Gaffney, J. S., Marley, N. A., Laskin, A., Shutthanandan, V., Xie, Y., Brune, W., Lesher, R., Shirley, T., and Jimenez, J. L.: Characterization of ambient aerosols in Mexico City during the MCMA-2003 campaign with Aerosol Mass Spectrometry: results from the CENICA Supersite, Atmos Chem Phys, 6, 925-946, 2006. Farmer, D. K., Matsunaga, A., Docherty, K. S., Surratt, J. D., Seinfeld, J. H., Ziemann, P. J., and Jimenez, J. L.: Response of an aerosol mass spectrometer to organonitrates and organosulfates and implications for atmospheric chemistry, P Natl Acad Sci USA, 107, 6670-6675, DOI 10.1073/pnas.0912340107, 2010. Kuwata, M., Zorn, S. R., and Martin, S. T.: Using Elemental Ratios to Predict the Density of Organic Material Composed of Carbon, Hydrogen, and Oxygen, Environ Sci Technol, 46, 787–794, 10.1021/es202525q, 2012. Peng, Z., Day, D. A., Stark, H., Li, R., Lee-Taylor, J., Palm, B. B., Brune, W. H., and Jimenez, J. L.: HOx radical chemistry in oxidation flow reactors with low-pressure mercury lamps systematically examined by modeling, Atmos. Meas. Tech., 8, 4863-4890, 10.5194/amt-8-4863-2015, 2015. Palm, B. B., Campuzano-Jost, P., Ortega, A. M., Day, D. A., Kaser, L., Jud, W., Karl, T., Hansel, A., Hunter, J. F., Cross, E. S., Kroll, J. H., Peng, Z., Brune, W. H., and Jimenez, J. L.: In situ secondary organic aerosol formation from ambient pine forest air using an oxidation flow reactor, Atmos. Chem. Phys., 16, 2943-2970, 10.5194/acp-16-2943-

2016, 2016. Fry, J. L., Brown, S. S., Middlebrook, A. M., Edwards, P. M., Campuzano-Jost, P., Day, D. A., Jimenez, J. L., Allen, H. M., Ryerson, T. B., Pollack, I., Graus, M., Warneke, C., de Gouw, J. A., Brock, C. A., Gilman, J., Lerner, B. M., Dubé, W. P., Liao, J., and Welti, A.: Secondary organic aerosol (SOA) yields from NO3 radical + isoprene based on nighttime aircraft power plant plume transects, Atmos. Chem. Phys., 18, 11663-11682, 10.5194/acp-18-11663-2018, 2018. Nault, B. A., Campuzano-Jost, P., Day, D. A., Schroder, J. C., Anderson, B., Beyersdorf, A. J., Blake, D. R., Brune, W. H., Choi, Y., Corr, C. A., de Gouw, J. A., Dibb, J., DiGangi, J. P., Diskin, G. S., Fried, A., Huey, L. G., Kim, M. J., Knote, C. J., Lamb, K. D., Lee, T., Park, T., Pusede, S. E., Scheuer, E., Thornhill, K. L., Woo, J. H., and Jimenez, J. L.: Secondary Organic Aerosol Production from Local Emissions Dominates the Organic Aerosol Budget over Seoul, South Korea, during KORUS-AQ, Atmos. Chem. Phys. Discuss., 2018, 1-69, 10.5194/acp-2018-838, 2018. Palm, B. B., de Sá, S. S., Day, D. A., Campuzano-Jost, P., Hu, W., Seco, R., Sjostedt, S. J., Park, J. H., Guenther, A. B., Kim, S., Brito, J., Wurm, F., Artaxo, P., Thalman, R., Wang, J., Yee, L. D., Wernis, R., Isaacman-VanWertz, G., Goldstein, A. H., Liu, Y., Springston, S. R., Souza, R., Newburn, M. K., Alexander, M. L., Martin, S. T., and Jimenez, J. L.: Secondary organic aerosol formation from ambient air in an oxidation flow reactor in central Amazonia, Atmos. Chem. Phys., 18, 467-493, 10.5194/acp-18-467-2018, 2018.

---

## Author Comment (AC1) · 3 Jan 2019

**Responses to Referee #1's comments**

**General comment:** In this manuscript, the authors report the first measurements of the rate constant for the reaction of eugenol, an atmospherically abundant methoxyphenol, with the hydroxyl radical in the gas phase. The results are placed in the context of other previously investigated methoxyphenols, including a helpful discussion of substituent effects. The authors also present a detailed characterization of the SOA yield and its response to $SO_2$ and $NO_2$, including a surprising enhancement in SOA yield due to the presence of $NO_2$, which may apply to other methoxyphenols. The experimental work is thorough and precise, with appropriate controls, and the manuscript is well organized. I recommend the manuscript for publication following minor revisions.

**Response to comment:** Many thanks for the reviewer's constructive comments and valuable suggestions, which would be much helpful to improve the scientific merits of this manuscript. The concerns raised by the reviewer have been carefully addressed in the revised manuscript.

**Comment 1:** 144: Though photolysis does not contribute to the decay of eugenol, could it contribute to the evolution of the oxidation products, either in the gas or particle phases? Though it is not a focus of this study (one can imagine forming SOA in one OFR, scrubbing any remaining ozone, and then irradiating the products in a second OFR), perhaps the possibility of photolysis of the oxidation products should be acknowledged at the end of this paragraph.

**Response to comment 1:** Thank you for your constructive suggestion. As you pointed out that it is unclear that how UV lights affect the evolution of oxidation products. Using a second OFR, this question can be answered. Unfortunately, we do not have a second OFR in our laboratory. This will be investigated in the future. According to your valuable suggestion, the discussion about the possibility of photolysis of the oxidation products in the OFR has been added in the revised manuscript.

**Revision in the manuscript:**

**Lines 154-155, Add:** "However, it cannot be ruled out that photolysis under UV irradiation might have influence on the evolution of the oxidation products."

**Comment 2:** 164: I enjoyed the comparison of the experimental and AOP WIN-predicted rate constants. I wonder if the comparison should be placed in the context of other methods of prediction. For example, the DFT-predicted rate constant for the reaction of guaiacol with the hydroxyl radical (DOI: 10.1002/poc.3713) is about 1.6 times greater than the experimental value (DOI: 10.1021/jp1071023). In this context, the present agreement, with a predicted value about 0.8 times the experimental value, seems quite good.

**Response to comment 2:** According to your constructive suggestion, the comparison of rate constant of guaiacol with OH radicals obtained by density functional theory (DFT) calculation and experiment study has been added in the revised manuscript.

**Revision in the manuscript:**

**Lines 178-183, Add:** "In addition, the difference between density functional theory (DFT) calculation and lab study has been also observed. For example, the DFT-predicted rate constant of 2-methoxyphenol with OH radicals ($12.19 \times 10^{-11}$ cm$^3$ molecule$^{-1}$ s$^{-1}$) is higher than that ($7.53 \times 10^{-11}$ cm$^3$ molecule$^{-1}$ s$^{-1}$) obtained by lab study (Coeur-Tourneur et al., 2010a; Priya and Lakshmipathi, 2017)"

**Comment 3:** 252: The lack of kinetic limitation to condensation is very interesting. Could this observation be related broadly to the viscosity of the SOA derived from eugenol and guaiacol? How does the present relative humidity of about 44% compare to that in the previous OFR and smog chamber experiments discussed in the comparisons?

**Response to comment 3:** The lack of kinetic limitation on SOA condensation might be mainly related to the physico-chemical properties of the SOA derived from the OH-initiated reactions of eugenol and guaiacol, such as viscosity, low volatility, and high oxidation state, etc.

The relative humidity (RH) in this work was similar to that in the previous OFR study about SOA formation from guaiacol oxidation by OH radicals, while was higher than that in the previous smog chamber experiments conducted by Lauraguais et al. (2014) and Yee et al. (2013).

**Revision in the manuscript:**

**Line 272, Add:** "conducted at low RH (Fig. S6) (Lauraguais et al., 2014b; Yee et al., 2013)"

**Comment 4:** 365: Perhaps the detailed discussion of the effects of $NO_2$ on the SOA yield would benefit from a mechanistic schematic.

**Response to comment 4:** Thank you very much for your valuable suggestion. In this work, the oxidation products could be not identified due to the lack of analytical instruments. Thus, it is very difficult to discuss the effect of $NO_2$ on SOA yield from the point of mechanistic schematic. According to the master chemical mechanism (MCM) of aromatic compounds, $NO_2$ has influence on not only $O_x/HO_x$ chemistry but also the formation of nitrophenols and organonitrates. Therefore, we generally mentioned that $NO_2$ participated in the OH reaction of eugenol, consequently producing N-containing products. Based on $NO^+/NO_2^+$ ratios measured by the HR-ToF-AMS, it is suggested that most of N-containing products are organic nitrates. Thus, the relative low volatility of these products could be favorable of SOA formation (Duporté et al., 2016; Liu et al., 2016).

In addition, the fraction of organic nitrates has been calculated to be in the range of 25.64% to 82.05% in the revised manuscript, using the $NO^+/NO_2^+$ ratios obtained at different OH exposure, according to the method described by Fry et al. (2013).

**Revision in the manuscript:**

**Lines 418-421, Add:** "According to the method described by Fry et al. (2013) (shown in Supplement), the fraction of organic nitrate was calculated to be in the range of 25.64% to 82.05%, using the $NO^+/NO_2^+$ ratios (3.98−6.09) obtained at different OH exposure."

**Comment 5:** 427: I agree that it is very likely that the SOA derived from eugenol is light-absorbing. Other products, in addition to those containing nitrogen, could conceivably contribute to the proposed absorptivity. For example, products of oligomerization like biphenyls have been observed in the aqueous oxidation of phenolic species (DOI:10.5194/acp-14-13801-2014), and this pathway is likely relevant in the highly-concentrated aerosol phase.

**Response to comment 5:** According to your valuable suggestion, the formation of oligomers via OH-initiated reaction of methoxyphenols (Yu et al., 2014) has been added in the revised manuscript, and the discussion about their light absorption has also been added.

**Revision in the manuscript:**

**Lines 460-463, Add:** "In addition, the formation of oligomers in particle phase via OH-initiated reaction of methoxyphenols, which has been observed in aquesous oxidation of phenolic species (Yu et al., 2014), might also enhance light absorption in UV-visible region."

**Comment 6:** Technical corrections:

77: In the phrase "a type of methoxyphenols", methoxyphenol should be singular. Perhaps this phrase is redundant and could be omitted.

115-116: Two instances of "approximate" should be "approximately".

126: "Vaccum" should be "vacuum".

199: In the caption to Figure S4, perhaps explain that the arrows indicate the maximum values (i.e, those listed in Table 2).

217-218: This phrase should be reworded to better reflect that the fragmentation occurs in the particles and that the products subsequently volatilize out of the particles.

238: This phrase is slightly confusing, since some measure of composition is determined using the AMS.

368: Should be "cyclic".

383: Should be "radicals".

414-415: This phrase should be reworded to give, for example, "more attention should be paid to SOA formation..."

442: Should be "formation".

**Response to comment 6:** Thank you very much, these technical errors have been corrected in the revised manuscript.

**Revisions in the manuscript:**

**Line 77, Delete:** "a type of methoxyphenols"

**Lines 116 and 118, Change:** "approximate" **To** "approximately"

**Line 129, Change:** "vaccum" **To** "vacuum"

**Supplement, Line 89, Add:** "The arrows indicate the maximum values (i.e., those listed in Table 2)."

**Lines 237-239, Change:** "which would generate a large amount of fragmented molecules that could not condense on aerosol particles" **To** "which would generate a large amount of fragmented molecules that subsequently volatilize out of aerosol particles"

**Line 258, Change:** "composition" **To** "product composition"

**Line 390, Change:** "yclic" **To** "cyclic"

**Line 410, Change:** "radicasls" **To** "radicals"

**Lines 442-443, Change:** "it should pay more attenion on the SOA formation from the OH oxidation of biomass burning emissions and its subsequent effect on haze evolution" **To** "more attention should be paid to the SOA formation from the OH oxidation of biomass burning emissions and its subsequent effect on haze evolution"

**Line 472, Change:** "foramtion" **To** "formation"

**Reference**

Duporté, G., Parshintsev, J., Barreira, L. M. F., Hartonen, K., Kulmala, M., and Riekkola, M.-L.: Nitrogen-containing low volatile compounds from pinonaldehyde-dimethylamine reaction in the atmosphere: A laboratory and field study, Environ. Sci. Technol., 50, 4693-4700, doi: 10.1021/acs.est.6b00270, 2016.

Fry, J. L., Draper, D. C., Zarzana, K. J., Campuzano-Jost, P., Day, D. A., Jimenez, J. L., Brown, S. S., Cohen, R. C., Kaser, L., Hansel, A., Cappellin, L., Karl, T., Hodzic Roux, A., Turnipseed, A., Cantrell, C., Lefer, B. L., and Grossberg, N.: Observations of gas- and aerosol-phase organic nitrates at BEACHON-RoMBAS 2011, Atmos. Chem. Phys., 13, 8585-8605, 10.5194/acp-13-8585-2013, 2013.

Lauraguais, A., Coeur-Tourneur, C., Cassez, A., Deboudt, K., Fourmentin, M., and Choel, M.: Atmospheric reactivity of hydroxyl radicals with guaiacol (2-methoxyphenol), a biomass burning emitted compound: Secondary organic aerosol formation and gas-phase oxidation products, Atmos. Environ., 86, 155-163, 10.1016/j.atmosenv.2013.11.074, 2014.

Liu, J., Lin, P., Laskin, A., Laskin, J., Kathmann, S. M., Wise, M., Caylor, R., Imholt, F., Selimovic, V., and Shilling, J. E.: Optical properties and aging of light-absorbing secondary organic aerosol, Atmos. Chem. Phys., 16, 12815-12827, doi: 10.5194/acp-16-12815-2016, 2016.

Yee, L. D., Kautzman, K. E., Loza, C. L., Schilling, K. A., Coggon, M. M., Chhabra, P. S., Chan, M. N., Chan, A. W. H., Hersey, S. P., Crounse, J. D., Wennberg, P. O., Flagan, R. C., and Seinfeld, J. H.: Secondary organic aerosol formation from biomass burning intermediates: Phenol and methoxyphenols, Atmos. Chem. Phys., 13, 8019-8043, 10.5194/acp-13-8019-2013, 2013.

Yu, L., Smith, J., Laskin, A., Anastasio, C., Laskin, J., and Zhang, Q.: Chemical characterization of SOA formed from aqueous-phase reactions of phenols with the triplet excited state of carbonyl and hydroxyl radical, Atmos. Chem. Phys., 14, 13801-13816, 10.5194/acp-14-13801-2014, 2014.

---

## Author Comment (AC2) · 3 Jan 2019

**Responses to Referee #2's comments**

**General comment:** This study investigated the OH reaction rate of pure eugenol compound and its SOA yield with a custom-built oxidation flow reactor (OFR). The impact of $NO_2$ and $SO_2$ influence on SOA formation was also investigated. The results of the study might be very interesting to many, yet quite a few items need to be clarified before it can be accepted for publication.

**Response to comment:** Many thanks for the reviewer's constructive comments and valuable suggestions, which would be much helpful to improve the scientific merits of this manuscript. The concerns raised by the reviewer have been carefully addressed in the revised manuscript.

**Comment 1:** Experiment design: The manuscript mainly deals with two topics: the rate constant of Eugenol-OH reaction and SOA yield of Eugenol. As for the 6 experiments listed in the Table 1, apart from the Eugenol, how about the combination of other species such like m-xylene/1,3,5-TMB/$SO_2$/$NO_2$. For experiments determines the rate constant for the reaction with OH, were $SO_2$/$NO_2$ added? And for SOA yield studies, what was the reference compound to derive OH? It seemed that here $SO_2$/m-xylene/1,3,5-TMB are all not appropriate to sever as the reference compound since m-xylene/1,3,5-TMB they themselves are also SOA precursors and $SO_2$ instead influence the SOA yield. It is quite confusing. The authors need to clarify in Table 1 what combinations (including reference compound for deriving OH) of species are prepared for obtaining the rate constant, and what combinations instead are for studying the SOA yield.

**Response to comment 1:** Thank you for your suggestion. When measuring the rate constant of eugenol-OH reaction, m-xylene or 1,3,5-TMB were used as reference in the absence of $SO_2$ and $NO_2$. This was pointed out in Table 1. When investigating SOA formation, both m-xylene and 1,3,5-TMB were not introduced into the reactor because they were also SOA precursors. The caption of Table 2 has been rewritten in the revised manuscript to make it more clear.

For SOA yield studies, OH exposure in the OFR was calculated using $SO_2$ as the

reference compound in the separate calibration experiments. The repeat experiments showed relative low uncertainties ($\pm 10.2\%$) about OH exposure. At the same time, OH exposure were also calculated using a box model (Peng et al., 2015). The modelled OH exposures were also in well agreement with the measured results. In order to investigate the effects of $NO_2$ and $SO_2$ on SOA formation, $NO_2$ and $SO_2$ were separately added in the OFR.

According to your valuable suggestion, more experimental conditions have been added in Table 2 in the revised manuscript, which is shown in Table R1.

**Table R1.** Experimental conditions and results for SOA formation.

| Expt. | [eugenol]$_0$[a] | $\Delta$[eugenol][b] | $M_0$[c] | $SO_2$ | $NO_2$ | $Y_{max}$[d] | OH Exposure[e] | $\tau$[f] |
|---|---|---|---|---|---|---|---|---|
| | ($\mu g\ m^{-3}$) | ($\mu g\ m^{-3}$) | ($\mu g\ m^{-3}$) | (ppbv) | (ppbv) | | ($10^{11}$ molecules $cm^{-3}$ s) | (d) |
| #1 | 272 | 265 | 29 | – | – | 0.11 | 5.41 | 4.17 |
| #2 | 351 | 339 | 54 | – | – | 0.16 | 5.41 | 4.17 |
| #3 | 485 | 474 | 83 | – | – | 0.18 | 5.41 | 4.17 |
| #4 | 636 | 625 | 145 | – | – | 0.23 | 5.41 | 4.17 |
| #5 | 874 | 858 | 241 | – | – | 0.28 | 7.37 | 5.68 |
| #6 | 1327 | 1304 | 399 | – | – | 0.31 | 8.91 | 6.87 |
| #7 | 273 | 267 | 40 | 41 | – | 0.15 | 5.41 | 4.17 |
| #8 | 273 | 266 | 35 | – | 40 | 0.13 | 5.41 | 4.17 |

[a] Initial eugenol concentrations.

[b] Reacted eugenol concentrations.

[c] SOA concentrations.

[d] Maximum SOA yields.

[e] Corresponding OH exposure of maximum SOA yields.

[f] Corresponding atmospheric aging time of maximum SOA yields, calculated using a typical [OH] in the atmosphere in this work ($1.5 \times 10^6$ molecules $cm^{-3}$) (Mao et al., 2009).

**Revision in the manuscript:**

Table R1 has been added in the revised manuscript (i.e., Table 2).

**Comment 2:** From the study only the overall rate constant for reaction with OH was obtained. It seemed that not so much degradation kinetics are presented as indicated by the title.

**Response to comment 2:** According to your valuable suggestion, "degradation kinetics"

in the title has been replaced by "rate constant".

**Revision in the manuscript:**

**Title, Change:** "Degradation kinetics and secondary organic aerosol formation from eugenol by hydroxyl radicals" **To** "Rate constant and secondary organic aerosol formation from the gas-phase reaction of eugenol with hydroxyl radicals"

**Comment 3:** Line 28: The enhancement values need not to have 4 significant figures.

**Response to comment 3:** Four significant figures of the enhancement values have been reduced to three significant figures.

**Revision in the manuscript:**

**Line 28, Change:** "38.57% and 19.17%" **To** "38.6% and 19.2%"

**Comment 4:** Lines 77-84: What's the overall concentration of eugenol in ambient air? How important does this precursor compared with other Methoxyphenols.

**Response to comment 4:** As a type of methoxyphenols, eugenol is a representative compound with a branched alkene group, which has been mentioned in the revised manuscript. Based on the previous work, the overall concentration of eugenol in ambient air is on the order of ng m$^{-3}$ and is comparable to the other methoxyphenols (Simpson et al., 2005; Bari et al., 2009). This has been added in the revised manuscript.

**Revision in the manuscript:**

**Lines 77-80, Add:** "4-Allyl-2-methoxyphenol (eugenol) is a typical methoxyphenol produced by ligin pyrolysis with a branched alkene group. It is widely detected in the atmosphere with the concentration on the order of ng m$^{-3}$, which is comparable to those of other methoxyphenols (e.g., guaiacol and syringol)"

**Comment 5:** Line 105: The [O$_3$] was in the range of 0.94-9.11 ppmv while you want to explore the reaction between OH radical and eugenol. Can you clarify whether such high level of O$_3$ was just used to generate OH radical in the mixing tube, or they indeed existed in the flow tube? If it is the latter case as your supplement material shows, then an evaluation of the interference from the O$_3$ is needed.

**Response to comment 5:** O$_3$ with the concentration of 0.94−9.11 ppmv in the OFR

was used to generate OH radicals, and its concentration decreased to 0.39−6.02 ppmv after the photo-reaction between $O_3$ and $H_2O$. In order to evaluate the possible decay of eugenol via the reaction with $O_3$ and the possible SOA formation from their reaction, the control experiments were conducted in this work. The results showed that the concentration of eugenol was not affected by $O_3$ and no SOA formation was observed by SMPS and HR-ToF-AMS. These results were mainly resulted from the short reaction time in the OFR and the low rate constants of $O_3$ with methoxyphenols ($\sim10^{-19}$ $cm^3$ $molecule^{-1}$ $s^{-1}$) (El Zein et al., 2015). These descriptions have been added in the Supplement.

The variations in the normalized concentrations of eugenol and reference compounds (i.e., 1,3,5-trimethylbenzene and *m*-xylene) in the presence of 9.11 ppmv $O_3$ are shown in Figure R1.

[Figure]

**Figure R1.** Variations in the normalized concentrations of eugenol and reference compounds (i.e., 1,3,5-trimethylbenzene and *m*-xylene) in the presence of 9.11 ppmv $O_3$.

**Revisions in the manuscript:**

**Lines 105-107, Change** "$O_3$ was produced by passing zero air through an $O_3$ generator (Model 610-220, Jelight Co., Inc.), and its concentration was in the range of 0.94−9.11 ppmv in this work measured with an $O_3$ analyzer (Model 205, 2B Technology Inc.)" **To** "$O_3$ with the concentration of 0.94−9.11 ppmv in the OFR was produced by passing zero air through an $O_3$ generator (Model 610-220, Jelight Co., Inc.), which was used to

produce OH radicals."

**Lines 140-142, Add:** "The possible effect of $O_3$ on the decay of eugenol and reference compounds was investigated in this work. As shown in Fig. S3, their concentrations were not affected by $O_3$. Meanwhile, no SOA formation was observed by the SMPS and HR-ToF-AMS."

**Supplement, Lines 24-34, Add:** "Before photochemical reaction, the concentration of $O_3$ in the OFR was in the range of 0.94−9.11 ppmv, which decreased to 0.39−6.02 ppmv due to the consumption by $H_2O$ with 254 nm UV light. In order to evaluate the possible decay of eugenol via the reaction with $O_3$ and the possible SOA formation from their reaction, the control experiments were conducted in this work. The results showed that the concentration of eugenol was not affected by $O_3$ and no SOA formation was observed by SMPS and HR-ToF-AMS. In addition, the possible effects of $O_3$ on the decay of reference compounds (i.e., 1,3,5-trimethylbenzene and *m*-xylene) were also investigated. The results showed that the decays of reference compounds by $O_3$ could be ignored in this work. The variations in the concentrations of eugenol and reference compounds (i.e., 1,3,5-trimethylbenzene and *m*-xylene) in the presence of 9.11 ppmv $O_3$ are shown in Fig. S3."

Figure R1 has been added in the revised Supplement.

**Comment 6:** Lines 109-110: This is not acceptable. The OH reactivity applied in this experiment is at least 80 s$^{-1}$ to 380 s$^{-1}$ with only Eugenol accounted (using OH reaction rate calculated in section 3.1), not mentioned the $SO_2$ (0-198 ppb) and $NO_2$ (0-109 ppb in line 27) added in the later experiment. It suggested the calculated OH exposure should be at least several times lower than the OH exposure calculated without considering the external OH reactivity ($=0$ s$^{-1}$ assumed in original calculation) (Peng et al., 2015)

**Response to comment 6:** Thank you very much for your valuable suggestion. In this work, the OH exposures calculated by $SO_2$ decay in separate calibration experiments were taken as the original exposures without the external OH reactivity.

The OH suppression by the external OH reactivity has been recalculated in the revised manuscript. According to the concentration of eugenol in this work, the OH

reactivity was in the range of about 85 s$^{-1}$ to 410 s$^{-1}$, calculated using the method described by Peng et al. (2015). Subsequently, according to the OFR exposure estimator (v2.3) developed by Jimenez's group based on the estimation equations reported in the previous work (Li et al., 2015; Peng et al., 2015, 2016), the maximum reduction of OH exposure by eugenol in the OFR is approximately 90%, which has been corrected in the revised manuscript. The detailed calculation has been added in the Supplement.

**Revisions in the manuscript:**

**Line 232, Change** "30%" **To** "90%"

**Lines 232-233, Add:** "Its detailed calculation has been shown in the Supplement."

**Supplement, Lines 59-74, Add:**

**3. Calculation of OH suppression**

The OH suppression by external OH reactivity in the OFR was estimated according to the OFR exposure estimator (v2.3) developed by Jimenez's group based on the estimation equations reported in the previous work (Li et al., 2015; Peng et al., 2015, 2016). The concentration of O$_3$ required by this estimator was in the range of 7−70 ppmv. Thus, O$_3$ with the concentrations of 7.8 and 9.1 ppmv in this work was used for this estimator. In addition, RH and rate constant for eugenol with OH radicals were 44% and $8.01 \times 10^{-11}$ cm$^3$ molecule$^{-1}$ s$^{-1}$ used in this estimator. The external OH reactivity in this estimator was only taken eugenol into account, due to its much higher concentration than those of SO$_2$ and NO$_2$. The external OH reactivity was calculated to be in the range of 85 s$^{-1}$ to 410 s$^{-1}$, according to the following equation (Peng et al., 2015):

$$OHR_{ext} = k_{eugenol+OH}[eugenol] \tag{S3}$$

where $OHR_{ext}$ is the external OH reactivity, $k_{eugenol+OH}$ is the rate constant of eugenol with OH radicals, and [eugenol] is the concentration of eugenol.

According to the parameters mentioned above, the maximum reduction of OH exposure by eugenol in the OFR was approximately 90%.

**Comment 7:** Lines 116-117: Better add the assumed average [OH] and the reference as well.

**Response to comment 7:** The assumed average OH concentration and the reference

have been added in the revised manuscript.

**Revision in the manuscript:**

**Lines 119-120, Add:** "which was calculated using a typical [OH] of $1.5 \times 10^6$ molecules cm$^{-3}$ in the atmosphere (Mao et al., 2009)."

**Comment 8:** Lines 124-125: How do the authors calibrate their aerodynamic size distribution in AMS? If the authors consider the chemical-composition based particle density (Kuwata et al., 2012; Salcedo et al., 2006), how about it when compared to the effective aerosol density applied here? The aerosol size distribution of each experiment should be considered separately. The 100% full cut size of AMS lens is around 600 nm in mobility size (Nault et al., 2018). The effective density calculation from dva/dm could be biased if the aerosol in OFR grow beyond the AMS lens cut.

**Response to comment 8:** In this work, the HR-ToF-AMS is size-calibrated using NH$_4$NO$_3$ particles with the diameter between 60−700 nm selected by a DMA. This sentence has been added in the revised manuscript.

According to the equation based on chemical composition (i.e., H/C and O/C) (Kuwata et al., 2012), the calculated particle density (1.7−2.1 g cm$^{-3}$) is higher than the effective particle density estimated using the equation $\rho=d_{va}/d_m$ (DeCarlo et al., 2004). The discrepancy might be mainly resulted from the relatively higher oxidation state of SOA in this work. In addition, we agree that aerosol size distribution of each experiment should be considered separately. But, assuming that particles were spherical and non-porous, all effective particle density was calculated to be in the range of 1.42 to 1.59 g cm$^{-3}$ in this work. Considering the insignificant discrepancy among all effective density, the average effective particle density of 1.5 g cm$^{-3}$ was applied. In addition, the maximum diameters of SOA for all experiments were lower than 600 nm. Thus, the bias of the effective particle density should be reasonably insignificant, calculated using the equation $\rho=d_{va}/d_m$ (DeCarlo et al., 2004).

**Revision in the manuscript:**

**Lines 131-132, Add:** "The particle size for HR-ToF-AMS measurement was calibrated using NH$_4$NO$_3$ particles with the diameter between 60−700 nm selected by a DMA."

**Comment 9:** Lines 152-153: The decay of $SO_2$ was used to calculate [OH], so why was $SO_2$ not used as the reference compound in the relative rate method? As mentioned above, the reader should be informed in Table 1 what were initially added and what are the reference compound.

**Response to comment 9:** In general, the reference compounds are selected to have OH rate constants similar in magnitude to that of the test compound (Edney et al., 1986). The rate constant of $SO_2$ with OH radicals is $9 \times 10^{-13}$ $cm^3$ molecule$^{-1}$ s$^{-1}$ (Davis et al., 1979), which is about 2 orders of magnitude lower than that of eugenol with OH radicals obtained by the relative-rate method in this work. Thus, $SO_2$ is not suitably selected as the reference compound in the relative-rate method.

According to your valuable suggestion, more experimental conditions have been added in Table 2 in the revised manuscript, which is shown in Table R1.

**Table R1.** Experimental conditions and results for SOA formation.

| Expt. | [eugenol]$_0$[a] | $\Delta$[eugenol][b] | $M_0$[c] | $SO_2$ | $NO_2$ | $Y_{max}$[d] | OH Exposure[e] | $\tau$[f] |
|---|---|---|---|---|---|---|---|---|
| | ($\mu$g m$^{-3}$) | ($\mu$g m$^{-3}$) | ($\mu$g m$^{-3}$) | (ppbv) | (ppbv) | | ($10^{11}$ molecules cm$^{-3}$ s) | (d) |
| #1 | 272 | 265 | 29 | – | – | 0.11 | 5.41 | 4.17 |
| #2 | 351 | 339 | 54 | – | – | 0.16 | 5.41 | 4.17 |
| #3 | 485 | 474 | 83 | – | – | 0.18 | 5.41 | 4.17 |
| #4 | 636 | 625 | 145 | – | – | 0.23 | 5.41 | 4.17 |
| #5 | 874 | 858 | 241 | – | – | 0.28 | 7.37 | 5.68 |
| #6 | 1327 | 1304 | 399 | – | – | 0.31 | 8.91 | 6.87 |
| #7 | 273 | 267 | 40 | 41 | – | 0.15 | 5.41 | 4.17 |
| #8 | 273 | 266 | 35 | – | 40 | 0.13 | 5.41 | 4.17 |

[a] Initial eugenol concentrations.

[b] Reacted eugenol concentrations.

[c] SOA concentrations.

[d] Maximum SOA yields.

[e] Corresponding OH exposure of maximum SOA yields.

[f] Corresponding atmospheric aging time of maximum SOA yields, calculated using a typical [OH] in the atmosphere in this work ($1.5 \times 10^6$ molecules cm$^{-3}$) (Mao et al., 2009).

**Revision in the manuscript:**

Table R1 has been added in the revised manuscript (i.e., Table 2).

**Comment 10:** Lines 135-137: This is confusing. Did the authors examine a full cycle of UV light applied in the experiment?

**Response to comment 10:** Yes. In order to investigate the possible photolysis of eugenol and reference compounds at 254 nm UV light in the OFR, the comparative experiments are performed with UV lamp turned on and turned off, when eugenol or reference compounds are introduced into the OFR. To describe accurately, this sentence has been supplemented in the revised manuscript.

**Revision in the manuscript:**

**Lines 144-146, Add:** "the comparative experiments were conducted with UV lamp turned on and turned off, when eugenol and reference compounds were introduced into the OFR"

**Comment 11:** Lines 140-141: What is the photon flux of 254 nm in OFR. How do the authors determine 254 exposure/OH exposure ratio?

**Response to comment 11:** The photo flux of 254 nm is $2.0 \times 10^{14}$ photon cm$^{-2}$ s$^{-1}$. According to the OH exposure calculated in this work, 254 nm photon flux/OH exposure is in the range of $1.6 \times 10^2$ to $1.7 \times 10^3$ cm s$^{-1}$, which has been added in the revised manuscript.

**Revision in the manuscript:**

**Line 151, Add:** "($1.6 \times 10^2$ to $1.7 \times 10^3$ cm s$^{-1}$)"

**Comment 12:** Lines 163-164: Is it possible that the difference between your measurement and the theory was caused by the O$_3$ reaction?

**Response to comment 12:** Considering that the decays of eugenol and reference compounds by O$_3$ was negligible, the difference between the measured rate constant and the theory rate constant was not reasonably caused by O$_3$. In addition, inaccurate performance of the AOP WIN model has been widely observed (Coeur-Tourneur et al., 2010; Lauraguais et al., 2012), because AOP WIN model is an empirical model (structure activity relationship model).

**Comment 13:** Lines 164-167: Have the authors considered the potential wall loss of

three species, which could result in different species decay ratios. Thus extra uncertainty on OH reaction rate coefficient of Eugenol could be introduced.

**Response to comment 13:** The possible wall losses of eugenol and reference compounds in the OFR in this work were investigated. But, the results showed that the insignificant wall loss (< 3%) was observed by HR-ToF-PTRMS. The uncertainty on the rate constant of eugenol with OH radicals has been shown in Table 1 in the original manuscript.

**Revisions in the manuscript:**

**Line 188, Change** "2.31" **To** "(2.31 $\pm$ 0.12)"

**Line 188, Change** "8.01" **To** "(8.01 $\pm$ 0.40)"

**Comment 14:** Lines 181-183: How about those reaction rate coefficients estimated from experiment when compared to those from the SAR method?

**Response to comment 14:** According to the US EPA AOP WIN model based on the structure activity relationship (SAR) (US EPA, 2012), the rate constants of the OH-initiated reactions of guaiacol, 2,6-dimethylphenol, and syringol are $2.98 \times 10^{-11}$, $5.05 \times 10^{-11}$, and $16.51 \times 10^{-11}$ cm$^3$ molecule$^{-1}$ s$^{-1}$, respectively, which have been added in the revised manuscript. Their corresponding rate constants obtained from experiments are $7.53 \times 10^{-11}$, $6.70 \times 10^{-11}$, and $9.66 \times 10^{-11}$ cm$^3$ molecule$^{-1}$ s$^{-1}$, respectively (Coeur-Tourneur et al., 2010; Thuner et al., 2004; Lauraguais et al., 2012). These differences among rate constants suggest that it is necessary to determine the rate constants of multifunctional organics through lab experiments.

**Revision in the manuscript:**

**Lines 198-202, Add:** "while their corresponding rate constants were calculated to be $2.98 \times 10^{-11}$, $5.04 \times 10^{-11}$, and $16.51 \times 10^{-11}$ cm$^3$ molecule$^{-1}$ s$^{-1}$, according to the US EPA AOP WIN model (US EPA, 2012). These differences among rate constants suggest that the rate constants of multifunctional organics should be necessarily determined via lab experiments"

**Comment 15:** Line 207: The decrease have also been reported in references of (Palm et al., 2016; Palm et al., 2018)

**Response to comment 15:** These two references have been added in the revised manuscript.

**Revision in the manuscript:**

**Lines 228, Add:** "Palm et al., 2016, 2018"

**Comment 16:** Line 211: Should be larger than 30% based on Fig. S4 in (Peng et al., 2015). Please show the detailed calculation results.

**Response to comment 16:** Thank you very much for your valuable suggestion. We are very sorry to make this mistake about OH suppression estimation. According to the concentration of eugenol in this work, the OH reactivity was in the range of about 85 $s^{-1}$ to 410 $s^{-1}$, calculated using the method described by Peng et al. (2015). Subsequently, according to the OFR exposure estimator (v2.3) developed by Jimenez's group based on the estimation equations reported in the previous work (Li et al., 2015; Peng et al., 2015, 2016), the maximum reduction of OH exposure by eugenol in the OFR was approximately 90%, which has been corrected in the revised manuscript. The detailed calculation has been added in the Supplement.

**Revisions in the manuscript:**

**Line 232, Change** "30%" **To** "90%"

**Lines 232-233, Add:** "Its detailed calculation has been shown in the Supplement."

**Supplement, Lines 59-74, Add:**

**3. Calculation of OH suppression**

The OH suppression by external OH reactivity in the OFR is estimated according to the OFR exposure estimator (v2.3) developed by Jimenez's group based on the estimation equations reported in the previous work (Li et al., 2015; Peng et al., 2015, 2016). The concentration of $O_3$ required by this estimator is in the range of 7−70 ppmv. Thus, $O_3$ with the concentrations of 7.8 and 9.1 ppmv in this work was used for this estimator. In addition, RH and rate constant for eugenol with OH radicals were 44% and $8.01 \times 10^{-11}$ $cm^3$ $molecule^{-1}$ $s^{-1}$ used in this estimator. The external OH reactivity in this estimator was only taken eugenol into account, due to its much higher concentration than those of $SO_2$ and $NO_2$. The external OH reactivity was calculated to be in the range of 85 $s^{-1}$ to 410 $s^{-1}$, according to the following equation (Peng et al., 2015):

$$OHR_{ext} = k_{eugenol+OH}[eugenol] \qquad\qquad (S3)$$

where $OHR_{ext}$ is the external OH reactivity, $k_{eugenol+OH}$ is the rate constant of eugenol with OH radicals, and [eugenol] is the concentration of eugenol.

According to the parameters mentioned above, the maximum reduction of OH exposure by eugenol in the OFR was approximately 90%.

**Comment 17:** Line 227: How about the wall loss of aerosols in the flow tube. The authors could examine the wall loss by directly injecting aerosols into the OFR.

**Response to comment 17:** According to our previous results reported by Liu et al. (2014), the wall loss of particles (< 3%) in this flow tube could be ignored, mainly resulted from the short residence time and a uniform velocity profile. The ignorable wall loss of particles in this flow tube has been added in the revised manuscript.

**Revision in the manuscript:**

**Lines 218-219, Add:** "The wall loss of aerosol particles in the OFR could be ignored, according to our previous results reported by Liu et al. (2014)."

**Comment 18:** Line 263: Have the author consider the $NH_4NO_3 \rightarrow CO_2$ effect, which could influence $f_{CO2}$ and thus O/C and H/C ratio substantially. This bias could be larger especially in the later $NO_2$ + Eugenol experiment.

**Response to comment 18:** Thank you very much for your valuable suggestion. Considering that zero air was used as the carrier gas in this work, the residual $NH_3$ could be ignored. In addition, the amount of ammonium salts formed in the OFR measured by HR-ToF-AMS was very low (< 0.2 μg m$^{-3}$). Thus, $NH_4NO_3$ should have insignificant influence on $f_{CO2}$ in this work.

**Comment 19:** Line 284-286: For saying this, OH exposure should be compared.

**Response to comment 19:** The comparison of OH exposure in smog chamber and PAM has been added in the revised manuscript.

**Revision in the manuscript:**

**Lines 305-307, Add:** "because OH exposure in the PAM reactor is about 1−3 orders of magnitude higher than that in smog chamber"

**Comment 20:** Line 294-296: We cannot definitely conclude that the decrease is due to the fragmented molecules formed through the oxidation of gas-phase species. So better add "probably" or "possibly" before "due to".

**Response to comment 20:** "Possibly" has been added before "due to" in the revised manuscript.

**Revision in the manuscript:**

**Line 316, Add:** "possibly"

**Comment 21:** Line 311: Can the formed $SO_4$ be fully explained by the $SO_2$ decay in $SO_2$+ Eugenol experiment?

**Response to comment 21:** According to the law of conservation of mass, the mass concentration of sulfate could be fully explained by the $SO_2$ decay.

For example, the sulfate concentration formed at the OH exposure of $12.55 \times 10^{11}$ molecules cm$^{-3}$ s was about 51 μg m$^{-3}$, shown in Figure 4 in the orignial manuscript. Meanwhile, the consumption of $SO_2$ by OH radicals was about 14 ppbv when the initial concentration of $SO_2$ was 41 ppbv $SO_2$. Therefore, the mass concentration of sulfate could be fully explained by the $SO_2$ decay, according to the law of conservation of mass.

**Comment 22:** Line 323: Why does the eugenol can partition quickly under acidic aerosol condition?

**Response to comment 22:** We are very sorry to make an expression mistake. This sentence has been rewritten in the revised manuscript.

**Revision in the manuscript:**

**Lines 345-347, Change** "Under acidic conditions, the gas-phase oxidation products of eugenol would be partitioned more quickly into the particle-phase and further oxidized into low volatility products, or produce oligomeric organics by acid-catalyzed heterogeneous reactions" **To** "Under acidic conditions, the gas-phase oxidation products of eugenol partitioned onto the particle-phase would be further oxidized into low volatility products or produce oligomers by acid-catalyzed heterogeneous reactions"

**Comment 23:** Lines 360-362: In the paper of Finewax et al. (2018), it is phenoxy radical rather than OH-aromatic adducts that react with $NO_2$ or $O_2$. In fact, the formations of phenoxy radical and OH-aromatic adduct from phenols are totally different in reaction pathways. Lines 366-367: Still, the reaction pathway through the $NO_2$ addition on the phenoxy radical was neglected by the author.

**Response to comment 23:** Thank you very much. According to your valuable suggestion, "OH-aromatic adduct" has been replaced by "phenoxy radical" in the revised manuscript.

**Revisions in the manuscript:**

**Lines 381 and 384, Change** "OH-aromatic adducts" **To** "phenoxy radicals"

**Line 388, Change** "OH-eugenol adduct" **To** "phenoxy radical"

**Comment 24:** Line 375: The authors could still specify the $NO_3$ exposure compared to OH exposure by assuming thermo-steady state of $NO_2$ and $O_3$.

**Response to comment 24:** According to your valuable suggestion, the $NO_3$ exposure was estimated using a box model. The maximum exposure of $NO_3$ radicals was calculated to be approximately $1.7 \times 10^{11}$ molecules $cm^{-3}$, using the maximum $O_3$ concentration of 9.11 ppmv in this work. This exposure was about one order of magnitude lower than the maximum OH exposure. In addition, the rate constant of $NO_3$ radicals with eugenol was reported to be $1.6 \times 10^{-13}$ $cm^3$ $molecule^{-1}$ $s^{-1}$ (Zhang et al., 2016), which is about 2 orders of magnitude lower than that ($8.01 \times 10^{-11}$ $cm^3$ $molecule^{-1}$ $s^{-1}$) for eugenol with OH radicals obtained in this work. Therefore, the decay of eugenol by $NO_3$ radicals was not predominant, compared to the reaction of eugenol with OH radicals.

Based on the discussion mentioned above, the detailed revision in the revised manuscript was pointed out as follows.

**Revision in the manuscript:**

**Lines 396-401, Add:** "Using the box model (Peng et al., 2015) and the maximum $O_3$ concentration (9.11 ppmv) in this work, the maximum $NO_3$ exposure was calculated to be approximately $1.7 \times 10^{11}$ molecules $cm^{-3}$ s. Compared to the rate constant of eugenol with OH radicals obtained in this work, the rate constant ($1.6 \times 10^{-13}$ $cm^3$ $molecule^{-1}$ $s^{-}$

[1]) of eugenol with NO$_3$ radicals was about 2 orders of magnitude lower (Zhang et al., 2016). Thus, the contribution of NO$_3$ radicals on the decay of eugenol was insignificant."

**Comment 25:** Line 391: In this method, if the NO$^+$ and NO$_2^+$ ions from organonitrate were missed, the organic nitrate calculation was underestimated (Farmer et al., 2010). The authors can use the real time NO$^+$/NO$_2^+$, and NO$^+$/NO$_2^+$ ratio from NH$_4$NO$_3$ and organonitrate (a factor of 2.25 ) (Fry et al., 2018) to calculate -ONO$_2$ group concentration for organonitrates.

**Response to comment 25:** The fraction of organic nitrates has been calculated to be in the range of 25.64% to 82.05% in the revised manuscript, using the NO$^+$/NO$_2^+$ ratios obtained at different OH exposure, according to the method described by Fry et al. (2013). In addition, the calculation method has been added in the revised Supplement.

**Revisions in the manuscript:**

**Lines 418-421, Add:** "According to the method described by Fry et al. (2013) (shown in Supplement), the fraction of organic nitrate was calculated to be in the range of 25.64% to 82.05%, using the $NO^+ / NO_2^+$ ratios (3.98−6.09) obtained at different OH exposure"

**Supplement, Lines 51-58, Add:**

**2. Calculation of organic nitrate fraction**

The fraction of organic nitrate can be typically calculated according to the following equation (Fry et al., 2013):

$$\text{RONO}_{2,\text{frac}} = \frac{(R_{\text{experiment}} - R_{\text{NH}_4\text{NO}_3})(1 + R_{\text{RONO}_2})}{(R_{\text{RONO}_2} - R_{\text{NH}_4\text{NO}_3})(1 + R_{\text{experiment}})} \tag{S2}$$

where $\text{RONO}_{2,\text{frac}}$ is the fraction of organic nitrate, $R_{\text{experiment}}$ is the ratio of $NO_2^+ / NO^+$ measured by HR-ToF-AMS in the experiments, $R_{\text{NH}_4\text{NO}_3}$ (0.295) and $R_{\text{RONO}_2}$ (0.13) are the $NO_2^+ / NO^+$ ratios for ammonium nitrate and organic nitrates, respectively (Fry et al., 2013).

**Comment 26:** Line 839: "ratioas" should be "ratio as".

**Response to comment 26:** "ratioas" has been replaced by "ratio as" in the revised manuscript.

**Revision in the manuscript:**

**Line 897, Change** "ratioas" **To** "ratio as"

**References**

[revised manuscript text omitted]